# Organotypic 3D Cell-Architecture Impacts the Expression Pattern of miRNAs–mRNAs Network in Breast Cancer SKBR3 Cells

**DOI:** 10.3390/ncrna9060066

**Published:** 2023-10-26

**Authors:** María de los Ángeles Gastélum-López, Maribel Aguilar-Medina, Cristina García Mata, Jorge López-Gutiérrez, Geovanni Romero-Quintana, Mercedes Bermúdez, Mariana Avendaño-Felix, César López-Camarillo, Carlos Pérez-Plascencia, Adriana S Beltrán, Rosalío Ramos-Payán

**Affiliations:** 1Faculty of Biological and Chemical Sciences, Autonomous University of Sinaloa, Josefa Ortiz de Domínguez s/n y Avenida de las Américas, Culiacan 80013, Sinaloa, Mexicomaribelaguilar@uas.edu.mx (M.A.-M.); geovanniromero@uas.edu.mx (G.R.-Q.); marianaavendano@uas.edu.mx (M.A.-F.); 2Faculty of Dentistry, Autonomous University of Chihuahua, Av. Escorza No. 900, Centro, Chihuahua 31125, Chihuahua, Mexico; mbermudez@uach.mx; 3Postgraduate in Genomic Sciences, Autonomous University of Mexico City, San Lorenzo 290, Col del Valle, Mexico City 03100, Mexico; cesar.lopez@uacm.edu.mx; 4National Cancer Institute, Av. San Fernando 22, Belisario Domínguez Sec. 16, Tlalpan, Mexico City 14080, Mexico; carlos.pplas@gmail.com; 5FES Iztacala, National Autonomous University of Mexico, Av. de los Barrios S/N, Los Reyes Ixtacala, Tlalnepantla 54090, Estado de Mexico, Mexico; 6Human Pluripotent Stem Cell Core, University of North Carolina at Chapel Hill, Chapel Hill, NC 27599, USA; adriana_beltran@med.unc.edu

**Keywords:** breast cancer, microRNAs, 2D cell culture, 3D cell culture, organotypic 3D cell culture

## Abstract

Background. Currently, most of the research on breast cancer has been carried out in conventional two-dimensional (2D) cell cultures due to its practical benefits, however, the three-dimensional (3D) cell culture is becoming the model of choice in cancer research because it allows cell–cell and cell–extracellular matrix (ECM) interactions, mimicking the native microenvironment of tumors in vivo. Methods. In this work, we evaluated the effect of 3D cell organization on the expression pattern of miRNAs (by Small-RNAseq) and mRNAs (by microarrays) in the breast cancer SKBR3 cell line and analyzed the biological processes and signaling pathways regulated by the differentially expressed protein-coding genes (DE-mRNAs) and miRNAs (DE-microRNAs) found in the organoids. Results. We obtained well-defined cell-aggregated organoids with a grape cluster-like morphology with a size up to 9.2 × 10^5^ μm^3^. The transcriptomic assays showed that cell growth in organoids significantly affected (all *p* < 0.01) the gene expression patterns of both miRNAs, and mRNAs, finding 20 upregulated and 19 downregulated DE-microRNAs, as well as 49 upregulated and 123 downregulated DE-mRNAs. In silico analysis showed that a subset of 11 upregulated DE-microRNAs target 70 downregulated DE-mRNAs. These genes are involved in 150 gene ontology (GO) biological processes such as regulation of cell morphogenesis, regulation of cell shape, regulation of canonical Wnt signaling pathway, morphogenesis of epithelium, regulation of cytoskeleton organization, as well as in the MAPK and AGE–RAGE signaling KEGG-pathways. Interestingly, hsa-mir-122-5p (Fold Change (FC) = 15.4), hsa-mir-369-3p (FC = 11.4), and hsa-mir-10b-5p (FC = 20.1) regulated up to 81% of the 70 downregulated DE-mRNAs. Conclusion. The organotypic 3D cell-organization architecture of breast cancer SKBR3 cells impacts the expression pattern of the miRNAs–mRNAs network mainly through overexpression of hsa-mir-122-5p, hsa-mir-369-3p, and hsa-mir-10b-5p. All these findings suggest that the interaction between cell–cell and cell–ECM as well as the change in the culture architecture impacts gene expression, and, therefore, support the pertinence of migrating breast cancer research from conventional cultures to 3D models.

## 1. Introduction

Breast cancer is the most common cancer around the world, affecting 2.26 million individuals and placing it in fifth place worldwide in 2020 [1]. In addition, Siegel et al. estimate that in the United States alone the number of new cases of breast cancer will be 300,590 and the estimated number of deaths will be 43,700 in 2023 [2]. Breast cancer exhibits a high degree of heterogeneity, encompassing distinct genotypic, phenotypic, and anatomical characteristics that exert profound influences on the clinical outcomes of patients [3]. Although the histological and molecular classifications of breast cancer may appear simplistic, they can predict biological behavior and facilitate the development of targeted therapeutic approaches [4]. Approximately 84 breast cancer cell lines have been categorized based on the presence of four major molecular subtypes: human epidermal growth factor receptor 2 (HER2)-enriched, basal-like, luminal A, and luminal B [5]. HER2-positive (HER2+) and triple-negative breast cancers (TNBC) exhibit the most unfavorable prognosis, primarily attributable to the intrinsic resistance or acquired insensitivity of certain HER2+ tumors to anti-HER2 therapies, as well as the unresponsiveness of TNBC to hormonal therapies or agents targeting HER2 receptors [6]. Consequently, extensive research efforts have been dedicated to the development of innovative therapeutic strategies. These approaches focus on either employing diverse combinations of drugs and treatment regimens that target multiple receptors or exploiting compensatory and downstream crosstalk signaling pathways associated with HER2 [7]. Interestingly, a large part of the available research on the HER2+ subtype has been performed using the SKRB3 cell line mostly cultured in monolayer with a conventional two-dimensional (2D) cell culture, this is due to its practical benefits and a huge amount of substantial information has been obtained until now; however, 2D culture has been gradually replaced, since cells grow up in a monolayer on a flat solid surface, lacking cell–cell and cell–extracellular matrix (ECM) interactions that can be found in native tumors, and acquiring artificial polarity, which may cause aberrant gene expression [8,9,10,11].

By contrast, 3D cell culture allows cancer cells to interact among themselves and with the ECM, closely mimicking the native microenvironment of tumors in vivo [12,13]. This results in the acquisition of morphological and cellular characteristics similar to the tumors in vivo [14], as well as the activation of cell signaling pathways leading to changes in expression of protein-coding genes (mRNAs) and non-coding genes such as microRNAs (miRNAs) [15,16]. For instance, lower cell proliferation, higher resistance to docetaxel and paclitaxel as well as changes in gene expression were shown in 3D cell culture compared with 2D in PC3, LNCaP, and DU145 prostate tumor cell lines [17]; moreover, in a report using these same cell lines, it was shown that integrin-mediated cell–ECM interactions can modulate tumor cell morphology as well as the expression of chemokine receptors which are associated with the invasive phenotype and progression of prostate cancer [18]. In a report, spheroids from the laryngeal carcinoma HLaC78 cell line showed upregulation of genes involved in cell adhesion and cell junctions, and downregulation of genes controlling cell cycle, DNA-replication, and DNA mismatch repair [19]. 

MicroRNAs are small RNA molecules of approximately 22 nucleotides that participate in the post-transcriptional downregulation of gene expression through interaction with the 3′ untranslated regions (3′UTR) of their target mRNA [20], resulting in mRNA degradation or translational repression into cytoplasmic P-bodies [21]. Nowadays, miRNAs have an important role in cancer since have been implicated in the initiation and progression of cancers as well as in chemotherapy resistance [21,22,23,24]. However, most of the studies reported on the expression of RNAs that code or do not code for proteins in cancer research have been derived from studies carried out in 2D cell cultures [25]. There are few studies of 3D cell culture-based global miRNA expression analysis of different types of cancer such as colorectal cancer [26], and basal and luminal breast cancer cells [27], as such, it is necessary to study the change of gene and miRNAs expression patterns in a 3D context to a better understanding of the cell signaling pathways of the HER2+ breast cancer. Therefore, in this study, a comparative analysis was conducted between the 3D culture and the conventional two-dimensional (2D) culture of HER2+ breast cancer SKRB3 cells to comprehensively assess the consequences of pattern changes in miRNA and mRNA expression on the putative cellular signaling pathways and associated biological processes. In our study, we obtained well-defined cell-aggregated organoids with a grape cluster-like morphology. In this discovery phase, the transcriptomic assays showed that cell growth in organoids significantly affected the expression patterns of miRNAs and mRNAs. In silico analysis showed that a subset of 11 upregulated DE-microRNAs target 70 downregulated DE-mRNAs involved in the regulation of cell morphogenesis, the regulation of cell shape, the regulation of the canonical Wnt signaling pathway, morphogenesis of the epithelium, and the regulation of cytoskeleton organization, as well as in the MAPK and AGE–RAGE signaling KEGG-pathways. The miRNA–mRNA interaction analysis showed that hsa-mir-122-5p, hsa-mir-369-3p, and hsa-mir-10b-5p are responsible for downregulating 81% of the 70 downregulated DE-mRNAs. 

## 2. Results

### 2.1. Organotypic 3D On-Top Cultures of SKBR3 form Grape Cluster-like Organoids

We tested the Geltrex On-Top and Embedded cultures to obtain SKRB3 organoids with cell–cell and cell–ECM interactions and used ultra-low attachment (ULA) as aggrupation control (Figure 1A). We found that, at 120 h of incubation, the organoids of the Embedded culture showed no significant differences in size, but were more numerous (*p* = 0.010), in comparison with those of ULA (Figure 1B,C). In contrast, the On-Top culture generated well-defined organoids with the biggest size and numbers (*p* < 0.001), and, therefore, was selected for the transcriptome analysis. 

The morphology of SKBR3 cells in conventional 2D culture has the typical epithelial shape, with a wide cytoplasm and well-organized actin fibers, observing lamellipodia and filopodia structures (Figure 2A,B). By contrast, in the 3D On-Top culture, cells had a rounded morphology with little cytoplasm, forming grape cluster-like organoids evidencing weak cell–cell contact (Figure 2C,D), and were up to 150 μm in size. Interestingly, discrete extensions of the cytoplasmic membrane in SKRB3 cells were observed, establishing direct connections with neighboring spheroid cells reminiscent of the tunneled nanotubes documented in prior studies involving 3D-cultured cancer cells [28]. These observations imply the existence of intricate cellular intercommunication mechanisms in 3D structures. In addition, the reconstruction allowed us to determine its size, reaching about 9.2 × 10^5^ μm^3^ (Figure 3A, Appendix A). Based on previously established classifications, SKBR3 organoids fall into the category of loosely aggregated cell clusters due to their low level of compaction [29]. The 3D reconstruction of the organoid shows similar zones to those identified by Kelly et al. [30], which consist of a core, a quiescent zone, and a proliferation zone (as shown in Figure 3B). However, further investigation is necessary to confirm whether these zones are indeed necrotic, quiescent, or proliferative.

### 2.2. 3D Culture Induces Important Changes in Small RNAs Expression in SKBR3 Cells

RNAseq analysis found different types of small RNAs present in the samples, 79.4% were identified as known miRNAs, 2.16% as novel miRNAs, 17.6% as piRNAs, 1.49% as snoRNAs, and 2.36% as tRNAs; therefore, we selected only the known miRNAs for further analysis. There were about 1320 known miRNAs in the samples (1359–1367 for 2D culture and 1322–1396 for 3D culture), of which 39 were differentially expressed (DE-microRNAs) with the established criteria (*p* < 0.01), 20 upregulated, and 19 downregulated (Table 1 and Figure 4A), which demonstrates that the cellular organization present in organoids, together with cell–ECM interactions, have a remarkable effect on gene expression.

Among the DE-microRNAs, those which were most upregulated are hsa-miR-410-3p (FC = 20.1), hsa-miR-6529-5p (FC = 15.9), hsa-miR-122-5p (FC = 15.4), hsa-miR-409-3p (FC = 12.2), and hsa-miR-369-3p (FC = 11.4). Among the DE-microRNAs, those which were most downregulated are hsa-miR-449c-3p (FC = −6.9), hsa-miR-449b-3p (FC = −5.7), hsa-miR-3689a-3p (FC = −5.3), hsa-miR-449a (FC = −4.2), and hsa-miR-1247-5p (FC = −3.9). We found that a set of 14 upregulated DE-microRNAs matched with 4370 experimentally validated microRNA-target interactions (Figure 5), while a set of 15 downregulated DE-microRNAs matched with 2547 gene targets (Figure 6).

### 2.3. The Expression Profile of mRNAs Is Downregulated under 3D Culture Conditions

On the other hand, the analysis of transcript expression, using Clariom D Assay human microarrays, showed 819 transcripts downregulated and 92 upregulated in the organoids compared to the conventional 2D culture of SKBR3 cells; of which, 172 have a gene annotation corresponding to known protein-coding genes or mRNAs (DE-mRNAs) and had the established criteria (adjusted *p* < 0.01), where 123 were downregulated and 49 were upregulated (Table 2 and Figure 4B), demonstrating, as in the case of the DE-microRNAs, the effect of the cellular organization present in the organoids, together with the cell–ECM interactions, on gene expression.

Among the DE-mRNAs, those which were most upregulated are SLC44A4 (FC = 6.7), TFF1 (FC = 5.5), BGN (FC = 3.9), PRODH (3.1), and SLC22A18 (2.9). Among the DE-mRNAs, those which were most downregulated are GLYATL2 (FC = −36.1), TGFB2 (FC = −16.1), DST (FC = −14.4), OLR1 (FC = −12.2), and TPR (FC = −12.0). Notably, signaling pathways such as FOXO and HIPPO are affected, and a decreased level of TGF-β and SMAD3 expression in the 3D-cultured cell line is present.

### 2.4. Upregulation of hsa-mir-122-5p, hsa-mir-369-3p, and hsa-mir-10b-5p Affects Most of the DE-mRNAs

The cross-matching between DE-microRNAs and DE-mRNAs found in the organoids derived from the SKBR3 cells showed that the magnitude of FC values was well correlated, since FC of upregulated DE-microRNAs were higher than the observed in the downregulated ones, and FC values of downregulated DE-mRNAs were higher than the upregulated ones. These observations were in alignment with the number of targeted genes from the sets of miRNAs, where 14 upregulated DE-microRNAs matched 4370 target genes, while 15 downregulated DE-microRNAs matched with 2547 targets.

A subset of 11 upregulated DE-microRNAs targets 70 of the 123 downregulated DE-mRNAs, giving a correlation of 57% (Table 3). These selected genes are involved in 150 biological processes, such as the regulation of morphogenesis of an epithelium, cell morphogenesis, cell shape, cytoskeleton organization, MAPK cascade, Ras protein signal transduction, and Wnt signaling, among others, as well as in the MAPK and AGE–RAGE signaling KEGG-pathways, with CCL2, WNT5A, NRP1, WNK1, and TGFB2 being involved in most of the processes (Table 4). Of note, hsa-mir-122-5p (FC = 15.4), hsa-mir-369-3p (FC = 11.4), and hsa-mir-10b-5p (FC = 20.1) regulated up to 81% of the 70 DE-mRNAs, highlighting their pivotal roles in downregulating genes, thus affecting the 3D architecture of organoids. Moreover, five downregulated DE-microRNAs regulated only five of the forty-nine upregulated DE-mRNAs, giving a correlation of 10% (Table 5), and for that reason, we cannot further study the grouped function of these genes.

## 3. Materials and Methods

### 3.1. Cell Culture

The human cell line SKBR3 (HTB-30. ATCC, Manassas, VA, USA), derived from a patient with breast carcinoma HER2+, was grown in Dulbecco’s modification of Eagle’s minimal medium (DMEM/F-12) supplemented with 10% fetal bovine serum (FBS) and 1% penicillin–streptomycin (Invitrogen, Waltham, MA, USA) at 37 °C, in a 5% CO_2_ atmosphere. Cells between passages 2 and 6 were used for all experiments.

### 3.2. 3D Cell Culture System

To obtain SKBR3 organoids, a standard On-Top protocol was used [4,27]; briefly, 24-well flat-bottom plates were coated with 150 μL of LDEV-free growth factor-reduced Geltrex (Gibco, USA) and incubated at 37 °C for 30 min to promote its gelation. Then, 4.2 × 10^4^ cells were seeded, and DMEM/F-12 supplemented medium was added to incubate for up to 5 days. The only modification was that the medium had no additional 5% Geltrex as reported. Wells without Geltrex were used as a control (2D), maintaining the rest of the conditions. Similarly, a standard Embedded (In-Gel) protocol was used; briefly, the same number of cells were mixed with Geltrex pregel and then placed in 24-well flat-bottom plates before gelation [31]. Additionally, single cells were diluted to a density of 5 × 10^3^ cells and placed into ultra-low attachment (ULA) 24-well plates and incubated under the same conditions to facilitate the formation of cell spheroids. Cells were monitored at 72, 96, and 120 h to measure the number and size of spheres using ImageJ software v 1.53t in which regions of interest were surrounded manually around the entire organoid from a bright-field microscopy image, and then the surface area was calculated in the ImageJ module (“Area Measurements of a Complex Object”).

### 3.3. Confocal Immunofluorescence Microscopy

Cells were fixed in 4% formaldehyde in 1 × PBS for 10 min at room temperature, and then were washed three times with 1 × PBS and permeabilized with 0.1% Triton X-100 for 10 min. Samples were washed and blocked with 22.5 mg/mL glycine and 1% bovine serum albumin, and then actin filaments were stained with 1:2000 Phalloidin-iFluor 488 (ab176753, Abcam, Waltham, MA, USA) for 5 min, washed, and mounted with a Fluoroshield with DAPI (Sigma-Aldrich, St. Louis, MO, USA). Visualization was performed at a 40× magnification using a Leica LSCM (TCS SP8, Leica Microsystems, Wetzlar, Germany).

### 3.4. RNA Isolation

RNA was extracted using Trizol reagent (Invitrogen) following the manufacturer’s instructions. The quantity and quality of total RNA were evaluated using Nanodrop (ThermoFisher Scientific, Waltham, MA, USA) and RNA integrity was assessed first using agarose gel electrophoresis at 60 V, stained with GelRed 0.5×, (Biotium, San Francisco, CA, USA) and visualized on a UV transilluminator. Additionally, it was evaluated by capillary gel electrophoresis using an RNA 6000 Nano Chip (Agilent 2100 Bioanalyzer, Agilent Technologies, Santa Clara, CA, USA). Samples with RNA integrity numbers ≥ 9.0 were used for these studies [32].

### 3.5. Small RNA Sequencing (RNAseq)

MicroRNA expression analysis of SKBR3 cells cultured in 2D and 3D culture was performed by whole Small RNAseq. The sequencing data obtained from this study has been deposited in the Gene Expression Omnibus (GEO) under Accession No. GSE239998 provided by NCBI, (NIH, Bethesda, MD, USA). The TruSeq Small RNA Library Prep kit (Illumina, San Diego, CA, USA) was used for the preparation of the library. The sequencing was performed with a total of 125 M reads for each sample. rRNAs were removed. Reads with Phred Quality Score values greater than 30 across all sequencing cycles were conserved. The filtered reads had a distribution of 6 to 36 bp in length, with a mean of 22 bp, which corresponds to the common size of miRNAs. Reads were aligned to the reference genome (GRCh38), miRBase v22.1, and non-coding RNA database (RNAcentral release 14.0) to classify the different types of small RNAs present in the samples. For the quantification of the miRNAs in the samples, Bowtie alignment mapping was performed in miRBase v21 and the counting of mature miRNAs was performed with the miRDeep2 Quantifier. 

To find the differentially expressed miRNAs (DE-microRNAs) between the experimental groups (3D vs. 2D), miRNAs were normalized using TMM (Trimmed mean of M-values), and Principal Component Analysis (PCA) was performed to assess the homogeneity of the experimental replicates. The detection of the DE-miRNAs was performed using Fold Change, and an exact test using edgeR per comparison pair. From the general list, miRNAs with a fold-change (FC) greater than 2 or less than -2 were filtered with significance values of *p* < 0.01.

### 3.6. Microarrays Hybridization and Analysis

Transcript expression analysis of SKBR3 cells cultured in both 2D and 3D conditions was conducted using Clariom D Assay human microarrays (GeneChip, Affymetrix, Singapore, CA, USA). The Microarray data obtained from this study has been deposited in the Gene Expression Omnibus (GEO) under Accession No. GSE239813 provided by NCBI. The hybridization for whole-genome transcriptome analysis was performed following the instructions provided by the manufacturer. Briefly, cDNA preparation and biotin labeling were carried out using the Affymetrix GeneChip WT Pico Kit. Subsequently, cRNA purification was performed using the Affymetrix magnetic bead protocol. Array processing was performed using the Affymetrix GeneChip™ Hybridization, Wash, and Stain Kit. The arrays were incubated for 16 h in an Affymetrix GeneChip 645 hybridization oven at 45 °C with rotation at 60 rpm. Fluorescence amplification was achieved by adding biotinylated anti-streptavidin and an additional aliquot of streptavidin–phycoerythrin stain. A confocal scanner (Affymetrix GeneChip Scanner 3000 7G plus) was utilized to capture the fluorescence signal at a resolution of 3 μm after excitation at 570 nm. The average signal from two sequential scans was calculated for each microarray.

For the subsequent analysis, Partek Genomic Suite v8.0 software was employed. All samples were normalized using the Robust Multiarray Average (RMA) method, which encompasses background correction, normalization, and calculation of expression values. Differential expression analysis was performed using one-way ANOVA. Differentially expressed protein-coding genes (DE-mRNAs) were selected between the groups based on an absolute fold-change of 2, and the Benjamini and Hochberg false discovery rate [23] was applied to account for multiple hypothesis testing. Genes with an adjusted *p*-value < 0.01 were considered significant.

### 3.7. In Silico Analysis

The TarBase v8.0 database was used to find experimentally validated target genes regulated by the DE-microRNAs. miRNet software v. 4.2 was used to define how many validated genes are regulated by each set of DE-microRNAs and to graph the regulation networks formed by them. The web-based portal Metascape v3.5 was used for pathway enrichment analysis of selected genes using the Kyoto Encyclopedia of Genes and Genomes (KEGG) and GO Biological Processes ontology sources.

### 3.8. Statistical Analysis

Experiments were performed in triplicate and results were represented as mean ± standard deviation (SD). Independent sample’s *t*-test was used to compare the means of both groups, considering *p* < 0.05 as statistically significant. Statistical analyses were performed using the software GraphPad v8.0.

## 4. Discussion

In the present study, we aimed to elucidate the effect of 3D organization cells and the presence of an ECM on the pattern of expression of miRNAs and mRNAs using a breast cancer HER2+ SKBR3 cell line. The most common 3D culture systems are based on the use of scaffolds, scaffold-free, and derived from tissues [31]. There are no studies evaluating differential expression patterns between On-Top, Embedded, and ULA cultures, nevertheless, it is expected that the presence of a scaffold could modify the gene expression of organoids. For instance, the first evidence that the ECM component, laminin regulates the gene expression and differentiation of the primary mammary cells through direct interaction with the cell integrins was seen in the 90s [33]. It is believed that modifications in the position of cells in the matrix can affect gene expression. Embedded cells have a strong interaction with the ECM but limited interaction with other cells, while On-Top cells have less interaction with the ECM they have better interactions with neighboring cells, which allows for organoid formation. Our research indicates that the On-Top culture resulted in well-defined organoids with larger sizes and greater numbers compared to Embedded cells (as shown in Figure 1A). Although ULA culture is typically used for cell aggregation, we used it as a 3D control culture because it lacks scaffolding and cell–ECM interactions, unlike the Geltrex matrix [34].

In the conventional 2D cell culture of SKBR3, we obtained the common cell morphology in clusters with a great amount of free-floating cells or very loosely attached round cells (Figure 2A,B), which have been previously reported [35,36]. However, this changed when the cells were grown in the 3D On-Top cell culture (Figure 2C,D), where the cells turned to a cell morphology frankly rounded, smaller in size with less cytoplasm and a grape-like appearance distinguished by their poor cell–cell contacts. It has been seen that grape-like cells formed less-closely associated colonies with reduced cell–cell adhesion which could be a reflection of the acquisition of the ability to escape from their neighbors in the primary tumor over their evolution as they acquired the ability to metastasize [30]. These changes in cell morphology influenced by 3D cell culture have been previously reported in SKBR3 cells on Matrigel [30,37]. Our On-Top generated organoids, derived from cultures using Geltrex, present a larger volume of up to 150 µm, and a better definition than the organoids derived from the SKBR3 cultivated in Matrigel, maintaining grape-like appearance [30] and also presenting discrete extensions of the cytoplasmic membrane, establishing direct connections with neighboring spheroid cells reminiscent of the tunneled nanotubes documented in previous studies involving 3D-cultured cancer cells [29]. 

We next evaluated the miRNA expression in the organoids, finding 39 differentially expressed miRNAs (DE-microRNAs): 20 miRNAs upregulated and 19 downregulated (Table 1). Interestingly, none of these sets of dysregulated miRNAs have been associated previously with the 3D cell culture system in breast cancer [27,38]. However, some of these downregulated DE-microRNAs, such as hsa-miR-449a, hsa-miR-4739, hsa-miR-449a, hsa-miR-34c-5p, hsa-miR-219a-5p, hsa-miR-34c-5p, hsa-miR-34c-5p, hsa-miR-219a-5p, has-miR-5091, and has-miR-943 have been previously associated with poor breast cancer prognosis [39], among other processes [40,41,42,43,44,45]. In addition, the upregulated miRNAs hsa-miR-127-3p, hsa-miR-223-3p, has-miR-4458, hsa-miR-10b-5p, hsa-miR-381-3p, has-miR-451a, hsa-miR-142-5p, has-miR-1246, hsa-miR-375-3p, and hsa-miR-4739, have been associated with several breast cancer processes, such as malignancy [40,46], tumor progression [41], and poor prognosis [47], as well as, cancer hallmarks’ activation as proliferation [48,49], apoptosis [49] and metastasis [49,50,51], among others [51,52,53,54,55,56]. 

In two previous reports from our group, using the triple-negative breast cancer (TNBC) cell line Hs578T and the luminal B breast cancer cell line BT-474 cultivated under the same cell culture conditions used here, we also obtained compact and large organoids, but with different morphology [4,27]. In the case of the BT-474 cell line, it has been shown that a 3D microenvironment can reprogram the oncogenic lncRNAs/mRNAs coexpression networks [4]. Similarly, in our case, SKBR3 cells reprogram the oncogenic miRNA/mRNA coexpresion network, which makes it obvious that lncRNA expression is also modified in our HER2+ subtype.

In the case of both basal Hs578T and luminal T47D breast cancer cell lines, the adoption of 3D organotypic cultures resulted in notable morphological alterations, including perturbations in cell–cell and cell–ECM interactions, the loss of cellular polarity, the reorganization of bulk structures, and the downregulation of specific miRNAs in Hs578T 3D cultures when compared to the 2D condition. This contributes to the positive modulation of crucial biological processes, such as the cellular response to hypoxia and focal adhesion, whereas the upregulation of miRNAs is associated with negative regulation of the cell cycle [20]. Meanwhile, in SKBR3 cells, in silico analysis showed that a subset of 11 upregulated DE-microRNAs target 70 downregulated DE-mRNAs involved in the regulation of cell morphogenesis, the regulation of cell shape, the regulation of canonical Wnt signaling pathway, the morphogenesis of epithelium, and the regulation of cytoskeleton organization, as well as in the MAPK and AGE–RAGE signaling KEGG-pathways. These data showed the impact of the cell culture system in each breast cancer subtype, in fact, in these reports, we did not find any deregulated DE-microRNAs that were observed in this work.

We also found, through the expression profile of mRNAs, that most of the genes are downregulated under 3D-cultured conditions. Interestingly, within the differentially expressed protein-coding genes (DE-mRNAs), we found genes involved in the tumoral microenvironment, such as CCL2, a chemokine involved in the tumoral progression of various cancers by modulating the tumor microenvironment [57], promoting cellular growth, migration, angiogenesis, and the recruitment of immunosuppressive cells through its activation at different stages of tumorigenesis. Additionally, its involvement as an autocrine or paracrine growth factor have been described [58,59]. WNT5A belongs to the Wnt signaling pathway family, which modulates different crucial processes for normal cellular development, including cell proliferation, adhesion, migration, and differentiation [60]. Loss of Wnt5 is associated with cancer relapse and poor survival, as it plays an important role as a tumor suppressor and inhibits cell migration by decreasing the production of matrix metalloproteinases (MMPs), contributing to cellular dispersion, reduced cell–collagen interaction, increased motility, and decreased adhesion [61,62]. Borcherding et al. reported that WNT5A is expressed in early breast cancer tumors, but as the tumor progresses to later stages and migrates to other tissues, its expression decreases [63]. NRP1 encodes neuropilins (NRPs), which are transmembrane glycoprotein receptors that act as co-receptors for vascular endothelial growth factor (VEGF) [64]. Overexpression of NRP1 has been reported in lung, colorectal, and breast cancer [65,66]. In breast cancer, its upregulation is identified as a tumor promoter, and its downregulation results in apoptosis promotion and inhibition of tumor growth [64]. PMEPA1 encodes the androgen-induced prostate transmembrane protein 1 and is highly regulated in prostate cancer [67], lung cancer [68], and breast cancer [69]. PMEPA1 negatively regulates TGF-β/SMAD signaling, thereby suppressing its tumor-suppressive capacity [70]. Previous work showed that the elimination of PMEPA1 in breast cancer cells significantly decreased their ability to form spheroids in Hs578t and BT-549 cells. Additionally, HCC1359 breast cancer cells lack phosphorylation levels of SMAD3, which correlate with decreased TMEPA1 expression [71]. This information is consistent with the decreased expression of TGF- β, SMAD3, and TMEPA1 in our 3D cultures. The dysregulation of all these genes constitutes a relevant pathway for the understanding and investigation of cancer biology.

One of the significant pathways implicated in the gene deregulation uncovered in our study is the Forkhead Box O (FOXO) pathway. The FOXO family of transcription factors exerts notable effects on cellular fate and tumor suppression across a broad spectrum of cancers, governed by stress and growth factors [72]. Reports have indicated that FOXO is downregulated in various types of cancers, including breast, colon, gastric, lung, and leukemias [73,74]. A prior study highlighted that the deletion of FOXO1 in adult mice heightens tumor incidence [75], while its activation halts the cell cycle and induces apoptosis in tumor cells [74]. This aligns with our findings, as evidenced by its downregulation in 3D cultures, reflecting closer concordance with in vivo observations. Another relevant gene in this pathway is HOMER1, which encodes a scaffold protein contributing to intracellular signal transduction [76], and is crucially engaged in Ca^2+^-dependent signaling [77]. As such, it has been reported to promote the proliferation, migration, and invasion in colorectal cancer cells via the G3BP1 pathway [78]. However, no reports of HOMER1’s involvement in breast cancer were found. In addition, an important finding is the HIPPO signaling pathway which is regarded as a tumor-suppressive pathway due to its pivotal role in regulating organ size, cell number, and tissue homeostasis [79]. Therefore, aberrant expression of some of its genes leads to increased cellular proliferation, tumorigenesis, and cancer metastasis [80,81]. This pathway encompasses mammalian sterile 20-like kinases 1/2 (MST1/2), large tumor suppressor kinases 1/2 (LATS1/2), Salvador 1 protein (SAV1), MOB kinases 1A/B (MOB1A/B), Yes-associated protein (YAP), and transcriptional coactivator with PDZ-binding motif (TAZ) [82]. Furthermore, the components of this pathway are reported to act as transducers, conferring cellular structural characteristics, polarity, shape, and cytoskeletal organization. These properties are closely associated with cells’ capacity to adhere to other cells and the extracellular matrix, and they are also influenced by the cellular microenvironment [81,82]. In our study, we observed that YAP1 expression is positively regulated in 2D-cultured conditions and decreased in 3D-cultured, consistent with findings from various research groups [83,84,85,86]. YAP activation occurs when cells grow on a rigid matrix and spread extensively, while YAP is inactivated when cells are seeded on a soft matrix and aggregate in a small area (round and compact geometry) [83,84,85]. Additionally, cell–cell contact, cellular geometry, matrix stiffness, and cell–matrix interactions [87,88] generate crucial signals that regulate the Hippo pathway. In low-density cell cultures, YAP is predominantly located in the nucleus, promoting proliferation. Conversely, when cell density is high, YAP is expressed in the cytoplasm, suppressing proliferation [84,86]. 

On the other hand, when the cross-matching between upregulated DE-miRNAs and downregulated DE-mRNA was performed, our results showed that 11 upregulated DE-microRNAs correlate with 70 downregulated DE-mRNAs (Table 3), which are involved in several biological processes and pathways (Table 4) that contribute to the development of breast cancer, such as Wnt [89,90] and MAPK [91] signaling. Additionally, the downregulated DE-mRNAs, such as TGFB2, DST, OLR1, GEN1, RASA2, FOXO1, JMJD1C, ROCK1, and WNT5A, have been previously associated with breast cancer as biomarkers [92,93,94,95] and therapeutic targets [96], or by participating in several processes such as metastasis [62,97,98], apoptosis [99], chemosensitivity [99], and epigenetic changes [100]. A highlighted downregulated DE-gene is WNT5A which in normal tissue belongs to Wnt/β-catenin-independent signaling, binding to different receptors to promote normal development in breast tissue [101,102,103]; its loss plays an important role in breast cancer progression [104]. This downregulated gene is a target for the upregulated hsa-mir-381-3p; its downregulation could be related to the overexpression of these miRNAs, which promote mammary carcinogenesis acting as oncomiRs. 

Notably, hsa-mir-122-5p (FC = 15.4), hsa-mir-369-3p (FC = 11.4), and hsa-mir-10b-5p (FC = 20.1) collectively controlled approximately 81% of the 70 DE-mRNAs. This underscores their crucial function in suppressing genes and consequently impacting the 3D structure of organoids. In the case of hsa-mir-369-3p it has been found in patients according to their lymph node status and associated with biological processes such as regulation of the epithelial-mesenchymal transition, cell proliferation, and transcriptional regulation [105]; it is downregulated in triple-negative breast cancer [106] and has been found in male breast cancer samples [107]. 

Hsa-mir-10b-5p has been extensively investigated in the context of breast cancer, particularly in comparison to other cancer types. Its upregulated expression has been consistently associated with various outcomes, including enhanced metastasis [108,109], increased invasive potential in both in vitro [110,111] and in vivo settings, augmented migration [112], elevated epithelial–mesenchymal transition [113], angiogenesis [114], and enhanced proliferation [112]. Collectively, these alterations contribute to unfavorable clinical outcomes, such as larger tumor size, advanced clinical stage, and shorter relapse-free survival periods [115,116,117,118]. Furthermore, hsa-mir-10b-5p exhibits associations with the expression of established biomarkers in breast cancer. Irrespective of metastatic status, hsa-mir-10b-5p expression positively correlates with HER2 positivity [117,119], while it negatively correlates with estrogen and progesterone receptor positivity [116,117]. This association further reinforces the connection between hsa-mir-10b-5p and the metastatic potential of breast cancer, as HER2 positivity and hormone receptor negativity serve as known predictors of tumor aggressiveness. Additional evidence supporting the pleiotropic effects of hsa-mir-10b-5p as a driving force in breast cancer invasiveness and metastasis comes from the observed positive correlation between hsa-mir-10b-5p expression and stemness, or self-renewal, in breast cancer stem cells [113]. Specifically, researchers discovered that stable overexpression of hsa-mir-10b-5p in MCF-7 cells led to heightened self-renewal capabilities and upregulation of genes associated with stemness and epithelial–mesenchymal transition. Conversely, the use of synthetic antagomirs against hsa-mir-10b-5p resulted in decreased self-renewal of stem cells [113].

Regarding hsa-mir-122-5p, it has a very important role in breast cancer. For instance, it has been shown that the downregulation of SDC1 mediated by hsa-mir-122-5p or liver-cell-derived exosomes would significantly augment the migratory capacity of breast cancer cells; furthermore, the metastatic potential or mobility of breast cancer cells is probably influenced by the presence of circulating hsa-mir-122-5p, which may not be directly correlated with the progression of breast cancer [120]. Fong et al. showed that elevated levels of miR-122 were observed in the blood of breast cancer patients with metastasis, indicating that cancer cells produce hsa-mir-122-5p, thereby facilitating metastasis through increased nutrient availability in the premetastatic niche. In vitro and in vivo experiments demonstrated that hsa-mir-122-5p derived from cancer cells inhibited glucose uptake by niche cells by suppressing pyruvate kinase, a key glycolytic enzyme. Notably, inhibiting hsa-mir-122-5p in vivo restored glucose absorption in distant organs such as the brain and lungs, thereby reducing the risk of metastasis. These findings suggest that extracellular hsa-mir-122-5p released by cancer cells can modulate systemic energy metabolism to drive disease progression by altering glucose utilization in recipient premetastatic niche cells [121]. It has also has been observed that, in TNBC cells, a substantial increase in hsa-mir-122-5p expression and a marked downregulation of the CHMP3 gene concluding that hsa-mir-122-5p, through the inhibition of CHMP3 via MAPK signaling, promotes aggressiveness and epithelial–mesenchymal transition (EMT) in TNBC [122]. Additionally, Wang et al. demonstrated, through modulation of the PI3K/Akt/mTOR/p70S6K pathway and targeting IGF1R, that hsa-mir-122-5p functions as a tumor suppressor and plays a crucial role in inhibiting the growth of new tumors. These findings indicate that hsa-mir-122-5p holds potential as a novel therapeutic or diagnostic/prognostic target for the treatment of breast cancer [123]. Finally, it has been shown that CDKN2B-AS1 regulates the expression of STK39 by acting as a sponge for hsa-mir-122-5p, thereby promoting breast cancer progression, and hsa-mir-122-5p modulates STK39 expression to regulate the impact of sh-CDKN2B-AS1 [42]. Interestingly, except for hsa-mir-10b-5p, none of these putative correlations have been previously reported in HER2+ breast cancer, opening a broad avenue of research to further understand the breast cancer biology using 3D cell culture experiments which are more approximate to the tumor microenvironment in vivo.

## 5. Conclusions

Our results show that the organotypic 3D cell-organization architecture of breast cancer SKBR3 cells impacts the expression pattern of the miRNAs–mRNAs network through overexpression of hsa-mir-122-5p, hsa-mir-369-3p, and hsa-mir-10b-5p. All these findings suggest that the interaction between the cell–cell and cell–extracellular matrix, as well as the change in the culture architecture impact gene expression, are pertinent to migrate breast cancer research from conventional cultures to 3D models.

## Figures and Tables

**Figure 1 ncrna-09-00066-f001:**
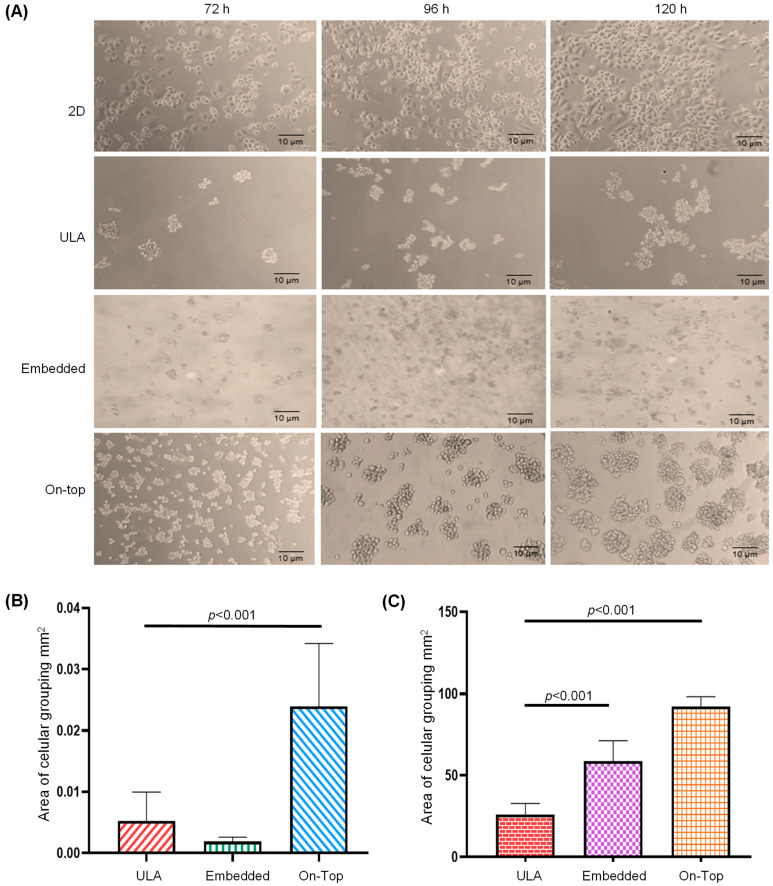
Formation and analysis of SKBR3 breast cancer cell organoids. Cell line was incubated for 72, 96, and 120 h under conventional 2D culture, ultra-low attachment (ULA), or with Geltrex in a 3D Embedded and On-Top culture (**A**). The area (mm^2^) (**B**) and number (**C**) of cell-aggregates were determined in 2D, ultra-low attachment (ULA), 3D Embedded, and On-Top cultures.

**Figure 2 ncrna-09-00066-f002:**
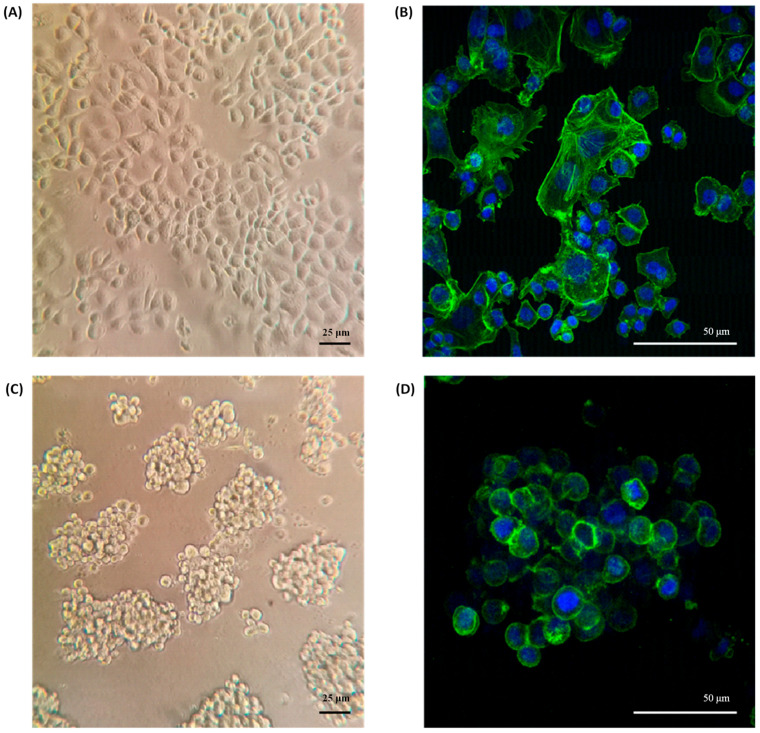
Morphology of SKBR3 organoids. (**A**) Conventional 2D culture on the plastic adherent surface observed by phase contrast microscopy at 200× magnification and (**B**) Confocal with maximum projection at 400× magnification. (**C**) Morphology of organoids grown on Geltrex by phase contrast and (**D**) confocal, labeling f-actin green and DAPI blue.

**Figure 3 ncrna-09-00066-f003:**
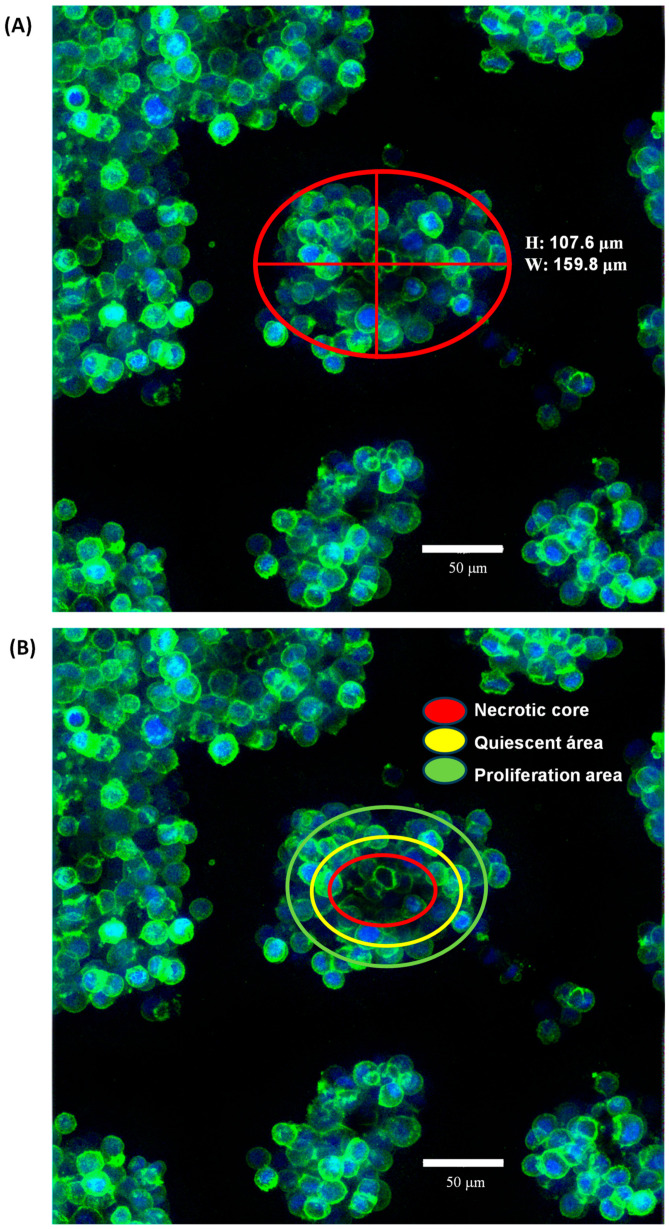
The morphology of SK-BR-3 organoids cultured using the On-Top method was analyzed at 120 h of cultivation through staining with dyes which enabled visualization of the actin cytoskeleton (phalloidin) and nuclei present in each organoid (DAPI) using a confocal microscope at a 40× objective. Based on previously established classifications by various authors, SK-BR-3 organoids fall into the category of loosely aggregated cell clusters due to their low level of compaction, with a morphology resembling clusters of grapes due to their poor cell–cell interaction (**A**). Nevertheless, the 3D reconstruction resembled the very well-established zones of the organoid that could be a necrotic core (red), a quiescent area (yellow), and a proliferation area (green). However, further investigation is necessary to confirm whether these zones are indeed necrotic, quiescent, or proliferative (**B**).

**Figure 4 ncrna-09-00066-f004:**
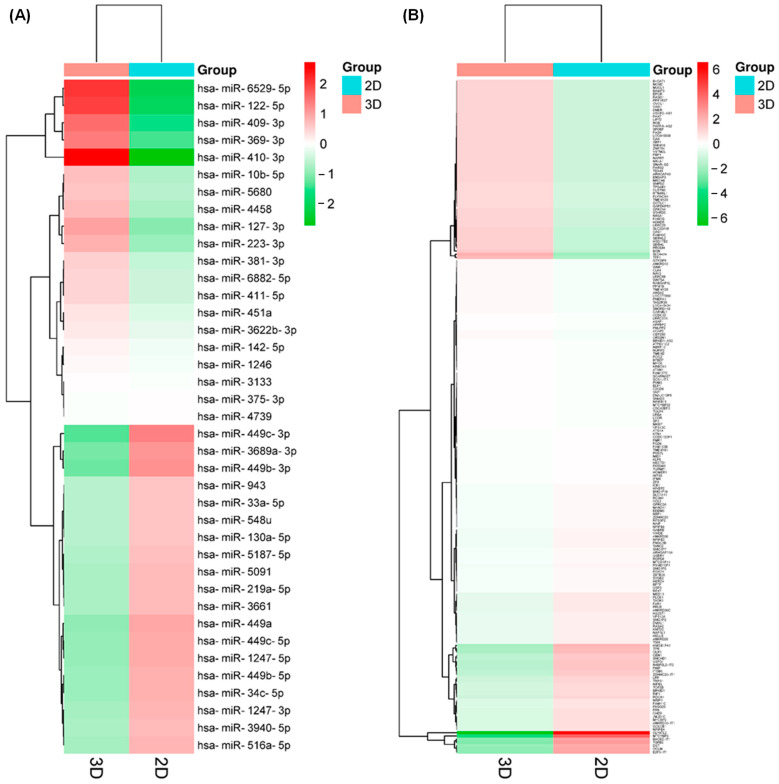
Differentially expressed protein-coding genes (DE-mRNAs) and miRNAs (DE-microRNAs) found in the organoids. Unsupervised hierarchical clustering analysis displaying the differential expression of DE-microRNAs (**A**) and DE-mRNAs (**B**) in both 2D and 3D cultures (Euclidean distance). Each column represents an individual sample, and each row represents a different miRNA. Red represents upregulation levels meanwhile green represents downregulation levels.

**Figure 5 ncrna-09-00066-f005:**
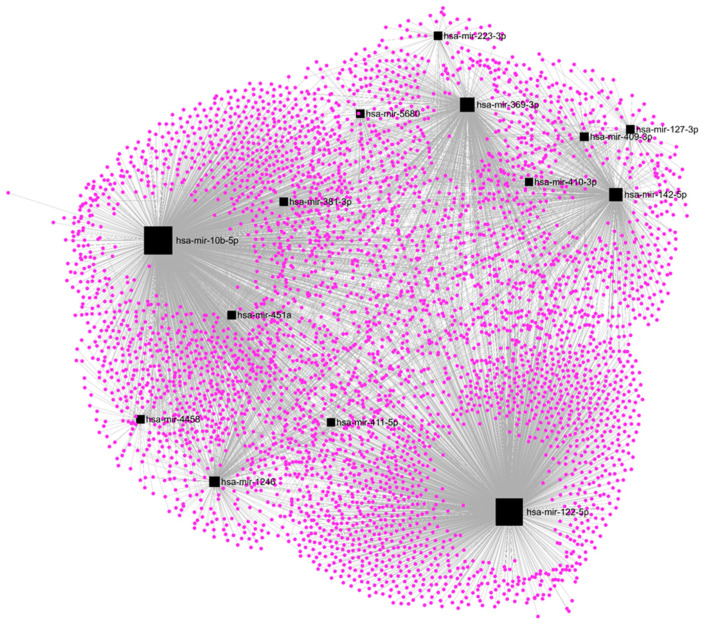
Regulatory networks among the set of upregulated DE-microRNAs and their experimentally validated target genes in SKBR3 organoids.

**Figure 6 ncrna-09-00066-f006:**
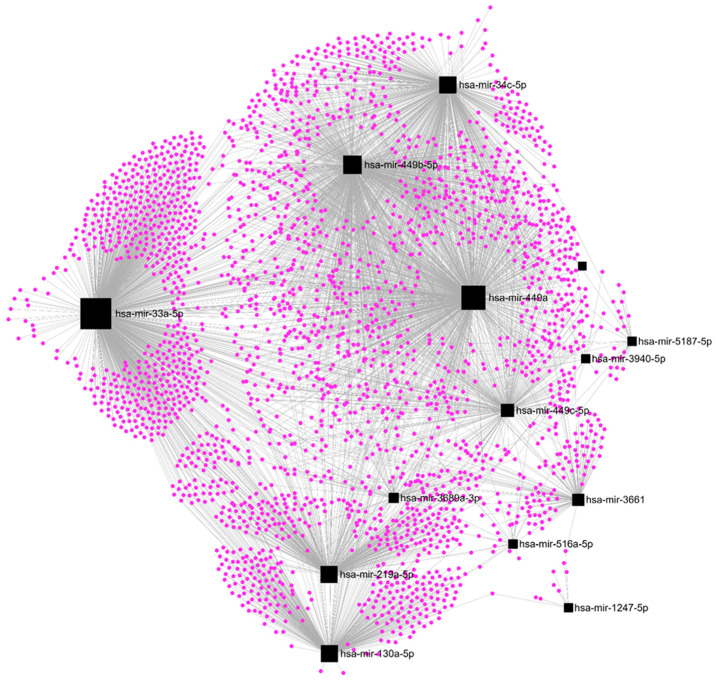
Regulatory networks among the set of downregulated DE-microRNAs and their experimentally validated target genes in SKBR3 organoids.

**Table 1 ncrna-09-00066-t001:** Differentially expressed up- and downregulated miRNAs (DE-microRNAs) found in SKBR3 organoids, in comparison to 2D culture.

DE-microRNAs Up	FC	*p*-Value	DE-microRNAs Down	FC	*p*-Value
hsa-miR-410-3p	20.1	0.0000	hsa-miR-449c-3p	−6.9	0.0000
hsa-miR-6529-5p	15.9	0.0000	hsa-miR-449b-3p	−5.7	0.0000
hsa-miR-122-5p	15.4	0.0000	hsa-miR-3689a-3p	−5.3	0.0071
hsa-miR-409-3p	12.2	0.0000	hsa-miR-449a	−4.2	0.0000
hsa-miR-369-3p	11.4	0.0000	hsa-miR-1247-5p	−3.9	0.0000
hsa-miR-127-3p	8.7	0.0000	hsa-miR-449c-5p	−3.8	0.0000
hsa-miR-223-3p	7.7	0.0054	hsa-miR-34c-5p	−3.6	0.0014
hsa-miR-4458	6.8	0.0005	hsa-miR-449b-5p	−3.5	0.0000
hsa-miR-10b-5p	6.5	0.0000	hsa-miR-1247-3p	−3.2	0.0000
hsa-miR-5680	6.4	0.0000	hsa-miR-516a-5p	−3.0	0.0002
hsa-miR-381-3p	5.4	0.0000	hsa-miR-3940-5p	−2.9	0.0057
hsa-miR-6882-5p	5.3	0.0100	hsa-miR-219a-5p	−2.7	0.0039
hsa-miR-411-5p	5.1	0.0000	hsa-miR-3661	−2.7	0.0007
hsa-miR-451a	4.0	0.0000	hsa-miR-5091	−2.6	0.0032
hsa-miR-3622b-3p	3.8	0.0004	hsa-miR-5187-5p	−2.4	0.0020
hsa-miR-142-5p	3.0	0.0000	hsa-miR-548u	−2.1	0.0004
hsa-miR-1246	2.6	0.0000	hsa-miR-130a-5p	−2.1	0.0017
hsa-miR-3133	2.1	0.0022	hsa-miR-943	−2.0	0.0053
hsa-miR-375-3p	2.0	0.0000	hsa-miR-33a-5p	−2.0	0.0000
hsa-miR-4739	2.0	0.0009			

**Table 2 ncrna-09-00066-t002:** Differentially expressed up- and downregulated protein-coding genes (DE-mRNAs) found in SKBR3 organoids, in comparison to 2D culture.

DE-mRNAs Up	FC	Adjusted *p*-Value	DE-mRNAs Down	FC	Adjusted *p*-Value	DE-mRNAs Down	FC	Adjusted *p*-Value
SLC44A4	6.7	0.0080	GLYATL2	−36.1	0.0013	ZDHHC20	−4.0	0.0081
TFF1	5.5	0.0006	TGFB2	−16.1	0.0097	NRP1	−4.0	0.0030
BGN	3.9	0.0054	DST	−14.4	0.0037	EDEM3	−4.0	0.0048
PRODH	3.1	0.0032	OLR1	−12.2	0.0052	SLC7A11	−3.9	0.0065
SLC22A18	2.9	0.0002	TPR	−12.0	0.0083	HIVEP2	−3.9	0.0076
OAS1	2.9	0.0023	USP34	−10.8	0.0019	ICE1	−3.9	0.0076
TENT5C	2.9	0.0046	PHIP	−10.6	0.0050	IPMK	−3.8	0.0078
SERHL2	2.8	0.0069	ITGB6	−10.3	0.0045	ZFX	−3.8	0.0092
HSD17B2	2.8	0.0007	GEN1	−10.0	0.0043	INTS6	−3.8	0.0043
GPKOW	2.6	0.0040	SMCHD1	−10.0	0.0078	HOMER1	−3.8	0.0078
STARD5	2.6	0.0038	TRPS1	−8.9	0.0071	HECTD1	−3.7	0.0044
NR5A1	2.6	0.0027	LPP	−8.8	0.0051	FAM133B	−3.6	0.0024
FANCG	2.6	0.0029	ROCK1	−8.5	0.0048	MID1	−3.5	0.0026
HDHD5	2.5	0.0038	RIF1	−8.4	0.0028	KLF8	−3.5	0.0035
LRRC26	2.5	0.0087	BRWD1	−8.3	0.0041	PUS7L	−3.5	0.0084
SMIM19	2.4	0.0028	TOP2B	−8.2	0.0002	TMEM181	−3.5	0.0036
ZNF764	2.4	0.0074	NIPBL	−8.1	0.0070	FMR1	−3.4	0.0038
VSTM2L	2.4	0.0086	MYCBP2	−7.8	0.0047	FGD6	−3.4	0.0085
LIPT2	2.3	0.0093	JMJD1C	−7.7	0.0074	KTN1	−3.3	0.0041
NGB	2.3	0.0014	GOLGB1	−7.6	0.0060	ATG14	−3.3	0.0052
SPDEF	2.3	0.0025	NPIPB4	−7.6	0.0078	LRBA	−3.2	0.0016
FA2H	2.3	0.0010	FRK	−7.3	0.0006	LCOR	−3.2	0.0067
GAA	2.3	0.0078	CHD9	−7.3	0.0074	SP3	−3.2	0.0082
XBP1	2.3	0.0050	NRIP1	−6.8	0.0025	MKI67	−3.2	0.0039
B4GAT1	2.2	0.0003	PRLR	−6.1	0.0049	VPS13C	−3.2	0.0093
MCM2	2.2	0.0045	ANKRD36C	−6.1	0.0002	SMAD3	−3.1	0.0006
MUCL1	2.2	0.0042	FAR1	−6.1	0.0023	NPIPB15	−3.1	0.0017
BAIAP3	2.2	0.0078	PLCE1	−6.0	0.0053	CDC42EP3	−3.1	0.0045
EPOR	2.2	0.0045	TAOK1	−6.0	0.0002	TULP4	−3.1	0.0021
RASD1	2.2	0.0046	MED13	−5.9	0.0059	PUM3	−3.0	0.0045
PPP1R37	2.2	0.0086	KMT2C	−5.7	0.0053	ELF1	−3.0	0.0040
OVOL1	2.2	0.0021	ANKRD28	−5.5	0.0039	C2CD5	−2.9	0.0015
VWA1	2.2	0.0035	NAP1L1	−5.4	0.0032	YAP1	−2.9	0.0057
DNER	2.2	0.0042	HELLS	−5.4	0.0054	BTBD7	−2.8	0.0048
PAX7	2.2	0.0051	HS2ST1	−5.3	0.0031	MYO6	−2.8	0.0012
FBP1	2.1	0.0016	DMXL1	−5.2	0.0025	KRBOX1	−2.8	0.0065
NAPRT	2.1	0.0008	RASA2	−5.2	0.0023	ATXN1	−2.8	0.0059
NR4A1	2.1	0.0066	VPS13A	−5.2	0.0011	ATP6V1C2	−2.7	0.0070
PARS2	2.1	0.0077	GABRE	−4.9	0.0028	NBPF10	−2.7	0.0040
TEX45	2.1	0.0079	FNDC3B	−4.8	0.0072	NUFIP2	−2.7	0.0050
ARHGAP40	2.1	0.0061	TANC2	−4.8	0.0033	CEMIP2	−2.7	0.0084
MROH6	2.1	0.0073	NPIPB3	−4.8	0.0019	POGZ	−2.7	0.0036
SMPD2	2.0	0.0040	VWDE	−4.7	0.0065	CCDC93	−2.6	0.0042
TPSAB1	2.0	0.0013	ANKRD36	−4.7	0.0071	LRRC37A	−2.6	0.0004
CLSTN3	2.0	0.0097	NPIPB2	−4.5	0.0001	ASAP1	−2.6	0.0093
RTN4RL1	2.0	0.0097	FOXO1	−4.5	0.0085	APPBP2	−2.6	0.0030
FLYWCH1	2.0	0.0007	ZBTB38	−4.5	0.0022	PHLPP2	−2.6	0.0019
TMEM129	2.0	0.0022	USP3	−4.4	0.0051	ACAP2	−2.5	0.0090
GGTLC1	2.0	0.0083	RFX7	−4.4	0.0005	CEP290	−2.5	0.0077
			BPTF	−4.4	0.0043	OR52N1	−2.5	0.0064
			HERC4	−4.4	0.0041	RABGAP1L	−2.4	0.0041
			SYDE2	−4.4	0.0080	PPM1B	−2.4	0.0094
			RGPD8	−4.3	0.0017	TMEM128	−2.4	0.0044
			QSER1	−4.3	0.0084	PMEPA1	−2.4	0.0063
			ARHGAP11A	−4.2	0.0075	ANKRD10	−2.3	0.0052
			GPRC5A	−4.1	0.0009	WNK1	−2.3	0.0016
			MAN2A1	−4.1	0.0077	CLK4	−2.3	0.0014
			CCL2	−4.1	0.0075	NAV2	−2.2	0.0018
			RC3H1	−4.1	0.0081	LRRC58	−2.2	0.0034
			NAIP	−4.0	0.0073	WNT5A	−2.2	0.0073
			NPIPB8	−4.0	0.0035	TAS2R30	−2.1	0.0092
						CARMIL1	−2.0	0.0096

**Table 3 ncrna-09-00066-t003:** Correlation between differentially expressed upregulated miRNAs (DE-microRNAs) and downregulated protein-coding genes (DE-mRNAs) found in SKBR3 organoids culture.

DE-microRNAs Up	DE-microRNAs Up	DE-microRNAs Up	DE-microRNAs Up	DE-mRNAs Down	FC	*p* Value	Description
hsa-mir-10b-5p				TGFB2	−16.1	0.0097	transforming growth factor beta 2
hsa-mir-122-5p	hsa-mir-369-3p	hsa-mir-10b-5p		DST	−14.4	0.0037	dystonin
hsa-mir-122-5p				OLR1	−12.2	0.0052	oxidized low density lipoprotein receptor 1
hsa-mir-122-5p				TPR	−12.0	0.0083	translocated promoter region, nuclear basket protein
hsa-mir-122-5p	hsa-mir-369-3p	hsa-mir-142-5p		USP34	−10.8	0.0019	ubiquitin specific peptidase 34
hsa-mir-369-3p				PHIP	−10.6	0.0050	pleckstrin homology domain interacting protein
hsa-mir-122-5p				GEN1	−10.0	0.0043	GEN1 Holliday junction 5′ flap endonuclease
hsa-mir-122-5p	hsa-mir-369-3p			SMCHD1	−10.0	0.0078	structural maintenance of chromosomes flexible hinge domain containing 1
hsa-mir-122-5p	hsa-mir-10b-5p	hsa-mir-142-5p		TRPS1	−8.9	0.0071	transcriptional repressor GATA binding 1
hsa-mir-122-5p	hsa-mir-369-3p			LPP	−8.8	0.0051	LIM domain containing preferred translocation partner in lipoma
hsa-mir-122-5p				RIF1	−8.4	0.0028	replication timing regulatory factor 1
hsa-mir-122-5p	hsa-mir-369-3p	hsa-mir-10b-5p	hsa-mir-1246	BRWD1	−8.3	0.0041	bromodomain and WD repeat domain containing 1
hsa-mir-451a				TOP2B	−8.2	0.0002	DNA topoisomerase II beta
hsa-mir-122-5p				NIPBL	−8.1	0.0070	NIPBL cohesin loading factor
hsa-mir-122-5p	hsa-mir-127-3p			MYCBP2	−7.8	0.0047	MYC binding protein 2
hsa-mir-122-5p	hsa-mir-369-3p			JMJD1C	−7.7	0.0074	jumonji domain containing 1C
hsa-mir-122-5p	hsa-mir-142-5p	hsa-mir-381-3p		CHD9	−7.3	0.0074	chromodomain helicase DNA binding protein 9
hsa-mir-122-5p				ANKRD36C	−6.1	0.0002	ankyrin repeat domain 36C
hsa-mir-369-3p	hsa-mir-4458			FAR1	−6.1	0.0023	fatty acyl-CoA reductase 1
hsa-mir-369-3p	hsa-mir-142-5p	hsa-mir-1246		TAOK1	−6.0	0.0002	TAO kinase 1
hsa-mir-1246				MED13	−5.9	0.0059	mediator complex subunit 13
hsa-mir-122-5p	hsa-mir-369-3p			KMT2C	−5.7	0.0053	lysine methyltransferase 2C
hsa-mir-122-5p				ANKRD28	−5.5	0.0039	ankyrin repeat domain 28
hsa-mir-369-3p				NAP1L1	−5.4	0.0032	nucleosome assembly protein 1 like 1
hsa-mir-122-5p				DMXL1	−5.2	0.0025	Dmx like 1
hsa-mir-369-3p	hsa-mir-10b-5p			RASA2	−5.2	0.0023	RAS p21 protein activator 2
hsa-mir-1246				VPS13A	−5.2	0.0011	vacuolar protein sorting 13 homolog A
hsa-mir-122-5p				GABRE	−4.9	0.0028	gamma-aminobutyric acid type A receptor subunit epsilon
hsa-mir-122-5p	hsa-mir-10b-5p	hsa-mir-142-5p		FNDC3B	−4.8	0.0072	fibronectin type III domain containing 3B
hsa-mir-122-5p				TANC2	−4.8	0.0033	tetratricopeptide repeat, ankyrin repeat and coiled-coil containing 2
hsa-mir-122-5p				VWDE	−4.7	0.0065	von Willebrand factor D and EGF domains
hsa-mir-10b-5p				ANKRD36	−4.7	0.0071	ankyrin repeat domain 36
hsa-mir-369-3p	hsa-mir-10b-5p	hsa-mir-223-3p		FOXO1	−4.5	0.0085	forkhead box O1
hsa-mir-381-3p				ZBTB38	−4.5	0.0022	zinc finger and BTB domain containing 38
hsa-mir-142-5p				RFX7	−4.4	0.0005	regulatory factor X7
hsa-mir-122-5p	hsa-mir-369-3p			HERC4	−4.4	0.0041	HECT and RLD domain containing E3 ubiquitin protein ligase 4
hsa-mir-122-5p				QSER1	−4.3	0.0084	glutamine and serine rich 1
hsa-mir-10b-5p	hsa-mir-381-3p			ARHGAP11A	−4.2	0.0075	Rho GTPase activating protein 11A
hsa-mir-10b-5p				CCL2	−4.1	0.0075	C-C motif chemokine ligand 2
hsa-mir-369-3p	hsa-mir-142-5p			RC3H1	−4.1	0.0081	ring finger and CCCH-type domains 1
hsa-mir-369-3p				ZDHHC20	−4.0	0.0081	zinc finger DHHC-type palmitoyltransferase 20
hsa-mir-381-3p				NRP1	−4.0	0.0030	neuropilin 1
hsa-mir-142-5p				EDEM3	−4.0	0.0048	ER degradation enhancing alpha-mannosidase like protein 3
hsa-mir-122-5p	hsa-mir-10b-5p			SLC7A11	−3.9	0.0065	solute carrier family 7 member 11
hsa-mir-10b-5p	hsa-mir-1246			HIVEP2	−3.9	0.0076	HIVEP zinc finger 2
hsa-mir-5680				IPMK	−3.8	0.0078	inositol polyphosphate multikinase
hsa-mir-127-3p				HECTD1	−3.7	0.0044	HECT domain E3 ubiquitin protein ligase 1
hsa-mir-122-5p				MID1	−3.5	0.0026	midline 1
hsa-mir-122-5p				KTN1	−3.3	0.0041	kinectin 1
hsa-mir-10b-5p				ATG14	−3.3	0.0052	autophagy related 14
hsa-mir-10b-5p	hsa-mir-142-5p			LCOR	−3.2	0.0067	ligand dependent nuclear receptor corepressor
hsa-mir-369-3p	hsa-mir-223-3p			SP3	−3.2	0.0082	Sp3 transcription factor
hsa-mir-10b-5p				MKI67	−3.2	0.0039	marker of proliferation Ki-67
hsa-mir-122-5p	hsa-mir-369-3p			VPS13C	−3.2	0.0093	vacuolar protein sorting 13 homolog C
hsa-mir-122-5p	hsa-mir-10b-5p			CDC42EP3	−3.1	0.0045	CDC42 effector protein 3
hsa-mir-1246				BTBD7	−2.8	0.0048	BTB domain containing 7
hsa-mir-10b-5p				MYO6	−2.8	0.0012	myosin VI
hsa-mir-122-5p				ATXN1	−2.8	0.0059	ataxin 1
hsa-mir-122-5p				NBPF10	−2.7	0.0040	NBPF member 10
hsa-mir-10b-5p				NUFIP2	−2.7	0.0050	nuclear FMR1 interacting protein 2
hsa-mir-1246				POGZ	−2.7	0.0036	pogo transposable element derived with ZNF domain
hsa-mir-142-5p				PHLPP2	−2.6	0.0019	PH domain and leucine rich repeat protein phosphatase 2
hsa-mir-369-3p				CEP290	−2.5	0.0077	centrosomal protein 290
hsa-mir-369-3p				RABGAP1L	−2.4	0.0041	RAB GTPase activating protein 1 like
hsa-mir-369-3p	hsa-mir-451a			PPM1B	−2.4	0.0094	protein phosphatase, Mg2+/Mn2+ dependent 1B
hsa-mir-122-5p				ANKRD10	−2.3	0.0052	ankyrin repeat domain 10
hsa-mir-122-5p	hsa-mir-1246			WNK1	−2.3	0.0016	WNK lysine deficient protein kinase 1
hsa-mir-122-5p				CLK4	−2.3	0.0014	CDC like kinase 4
hsa-mir-369-3p	hsa-mir-10b-5p			LRRC58	−2.2	0.0034	leucine rich repeat containing 58
hsa-mir-381-3p				WNT5A	−2.2	0.0073	Wnt family member 5A

**Table 4 ncrna-09-00066-t004:** Biological Processes and pathways in which correlated downregulated protein coding genes (DE-mRNAs) found in SKBR3 organoids participates.

Term	Description	DE-mRNAs Down
GO:0006325	Chromatin organization	NAP1L1, SMCHD1, RIF1, KMT2C, CHD9, JMJD1C
GO:0051276	Chromosome organization	TOP2B, POGZ, SMCHD1, NIPBL, RIF1
GO:0006281	DNA repair	POGZ, SMCHD1, NIPBL, RIF1, TAOK1, GEN1
GO:0048762	Mesenchymal cell differentiation	TGFB2, WNT5A, NRP1
GO:0002009	Morphogenesis of an epithelium	TGFB2, WNT5A, NRP1, HECTD1, BTBD7, CEP290
GO:0022604	Regulation of cell morphogenesis	CCL2, WNT5A, CDC42EP3, BRWD1, PHIP
GO:0008360	Regulation of cell shape	CCL2, CDC42EP3, BRWD1, PHIP
GO:0050921	Regulation of chemotaxis	WNT5A, NRP1, WNK1
GO:0051493	Regulation of cytoskeleton organization	MID1, TPR, NRP1, CDC42EP3, MYCBP2, TAOK1
GO:0002688	Regulation of leukocyte chemotaxis	CCL2, WNT5A, WNK1
GO:2000401	Regulation of lymphocyte migration	CCL2, WNT5A, WNK1
GO:0043408	Regulation of MAPK cascade	FOXO1, MID1, CCL2, TGFB2, WNT5A, NRP1, TAOK1
GO:0046578	Regulation of Ras protein signal transduction	RASA2, TGFB2, NRP1
GO:0030111	Regulation of Wnt signaling pathway	FOXO1, PPM1B, WNT5A, USP34, WNK1
GO:0070848	Response to growth factor	CCL2, TGFB2, TPR, WNT5A, NRP1
GO:0019827	Stem cell population maintenance	FOXO1, NIPBL, RIF1
KEGG:hsa04933	AGE-RAGE signaling pathway in diabetic complications	FOXO1, CCL2, TGFB2
KEGG:hsa04010	MAPK signaling pathway	PPM1B, RASA2, TGFB2, TAOK1

**Table 5 ncrna-09-00066-t005:** Correlation between differentially expressed downregulated miRNAs (DE-microRNAs) and upregulated protein coding genes (DE-mRNAs) found in SKBR3 organoids culture.

DE-microRNAs Down	DE-microRNAs Down	DE-microRNAs Down	DE-mRNA Up	FC	*p* Value	Description
hsa-mir-34c-5p	hsa-mir-449a		OAS1	2.9	0.0023	2′-5′-oligoadenylate synthetase 1
hsa-mir-449a	hsa-mir-449b-5p		STARD5	2.6	0.0038	StAR related lipid transfer domain containing 5
hsa-mir-34c-5p	hsa-mir-449a	hsa-mir-449b-5p	XBP1	2.3	0.0050	X-box binding protein 1
hsa-mir-3661			NAPRT	2.1	0.0008	nicotinate phosphoribosyltransferase
hsa-mir-449a			NR4A1	2.1	0.0066	nuclear receptor subfamily 4 group A member 1

## Data Availability

All data generated or analyzed during this study are included in this published article. The sequencing (Accession No. GSE239998) and Microarray (Accession No. GSE239813) data have been deposited in the Gene Expression Omnibus (GEO) under provided by NCBI, (NIH, Bethesda, MD, USA).

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
