# Peer review of "Organotypic 3D Cell-Architecture Impacts the Expression Pattern of miRNAs–mRNAs Network in Breast Cancer SKBR3 Cells"

_ncrna, 2023, doi:10.3390/ncrna9060066_

Round 1

Reviewer 1 Report

Significance: 

MS by María de los Ángeles Gastélum‐López et al. “Organotypic 3D cell‐architecture impacts the expression pattern of miRNAs‐mRNAs network in breast cancer SKBR3 cells” The question they are trying to answer is interesting and relevant in the ncRNA field but here are some points that the author can highlight in the MS.

 Authors use 3-D organoid culture to identify the differentially expressed miRNAs and mRNAs for breast cancer. The list of differentially expressed miRNAs identified a common set of tumor suppressor miRNAs but the authors did not mention the uniquely expressed miRNAs and mRNAs which reduce the impact of the study.

Author Response

Dear reviewer, we really appreciated your kind comments. All your recommendations have been followed and answered. Changes in the manuscript can be reviewed in the track changes of Word. Please find our response in the attached PDF file. We hope you find the paper now suitable for publication.

Best regards

Reviewer 2 Report

Organotypic 3D cell architecture impacts the expression pattern of miRNAs-mRNAs network in breast cancer SKBR3 cells" is aimed at studying the different expression patterns in the same cell line cultivated in both 2D and 3D environments. This topic is of significant interest in the scientific literature since 3D cultures closely resemble the human model. However, detailed explanations and clarifications are provided in the attached Word document.

Author Response

(The authors gave the same response as above.)

Reviewer 3 Report

Ramos-Payan and groups paper on the organotypic 3D cell architecture in breast cancer SKBR3 cells showing an impact on the regulatory RNA expression is an interesting study.

However, the major caveats are that it is performed in a single cell line of breast cancer, data from other cell lines have been published in other journals by the same group. There is no novelty from the perspective of experimental design or data that is not reflected in studies performed in other cancers and other cell lines by the same group.

2.       References 4, 20 are from the same group using breast cancer cells and evaluate the change in the regulatory-scape of different breast cancer cell lines. It begs the question, why is this study important with yet another different single cell line? a comparison of the results from the references 4 and 20 and interspersing the data of this paper would compare and contrast the findings and can be an excellent review. especially when comparing with other cancers.

3.       Why was SKBR3 cell line chosen? What is the significance? 

4.       What passage of SKBR3 cells was used in these experiments, have early/ late passages of cells been used in this study? Have they been compared for the changes in the expression of the RNA molecules evaluated. As cells age and or are being passaged in the labs, the gene expression changes. how is this controlled for?

5.       At the outset, figure 1 A. the cells from the embedded wells (3rd row) are of significantly poor quality when compared to the rest of the figures. This does not permit me to perform evaluations for size or number. This must be replaced to include better images the likeness of on-top or ULA.

6.       In results (pg 5 line 213. The claim is made that there was no significant differences in size was observed between the ULA and embedded. This may not be true if sufficient samples were selected? What was the N? how many times was this repeated?

 The sequencing data is just reporting of the findings and no validations or consequences have been determined or experimentally evaluated and hence I find the paper not sufficient enough to be accepted to ncRNA.

Consider revising some of the longer sentences to make them more concise.

Include a brief introductory paragraph to give readers an overview of what to expect.

Ensure consistency in the terminology used throughout the article- for example 3D and tridimensional has been used interchangeably after first use.  

Author Response

(The authors gave the same response as above.)

Round 2

Reviewer 1 Report

Thanks for the clarification.

Reviewer 3 Report

Dear Authors, 

Thank you for your responses. I feel that you have addressed my concerns near adequately. Although the arguments about lack of funding preventing the ability to perform functional validation is not valid or acceptable, I understand what is said. Albeit, for further publications, please consider incorporating the suggestions and making a complete rounded story that can be of interest. 

I accept these changes and would recommend to the editors that this paper be considered for publication (at the discretion of the editors).